# The Effect of Handlebar Height and Bicycle Frame Length on Muscular Activity during Cycling: A Pilot Study

**DOI:** 10.3390/ijerph19116590

**Published:** 2022-05-28

**Authors:** Ana Conceição, Vítor Milheiro, José A. Parraca, Fernando Rocha, Mário C. Espada, Fernando J. Santos, Hugo Louro

**Affiliations:** 1Department of Sport Sciences, Sport Sciences School of Rio Maior, 2040-413 Rio Maior, Portugal; vmilheiro@esdrm.ipsantarem.pt (V.M.); fernandorocha@esdrm.ipsantarem.pt (F.R.); hlouro@esdrm.ipsantarem.pt (H.L.); 2Research Centre in Sports, Health and Human Development (CIDESD), 5000-801 Vila Real, Portugal; 3Departamento de Desporto e Saúde, Escola de Saúde e Desenvolvimento Humano, Universidade de Évora, 7000-654 Évora, Portugal; jparraca@uevora.pt; 4Comprehensive Health Research Centre (CHRC), Universidade de Évora, 7000-654 Évora, Portugal; 5Polytechnic Institute of Setúbal, School of Education, 2914-504 Setúbal, Portugal; mario.espada@ese.ips.pt (M.C.E.); fernando.santos@ese.ips.pt (F.J.S.); 6Life Quality Research Centre, Complexo Andaluz, Apartado, 2040-413 Rio Maior, Portugal; 7Faculty of Human Kinetics, University of Lisbon, 1499-002 Cruz Quebrada, Portugal

**Keywords:** cycling, electromyography, handlebar height, bicycle frame

## Abstract

The cycling literature is filled with reports of electromyography (EMG) analyses for a better understanding of muscle function during cycling. This research is not just limited to performance, as the cyclist’s goal may be rehabilitation, recreation, or competition, so a bicycle that meets the rider’s needs is essential for a more efficient muscular activity. Therefore, the purpose of this study was to understand the contribution of the activity of each of the following muscles: TD (trapezius descending), LD (latissimus dorsi), GM (gluteus maximus), and AD (anterior deltoid) in response to different bicycle-rider systems (handlebar height; bicycle frame length) and intensities in a bicycle equipped with a potentiometer. Surface EMG signals from muscles on the right side of the body were measured. A general linear model test was used to analyze the differences between muscle activation in the test conditions. Effect sizes were calculated using a partial Eta2 (*η*^2^). The level of significance was set at 0.05. Muscle activation of different muscles differs, depending on the cycling condition (Pillai’s trace = 2.487; F (36.69) = 9.300; *p* < 0.001. *η*^2^ = 0.958), mostly during low intensities. In high intensities, one specific pattern emerges, with a greater contribution of GM and TD and weaker participation of LD and AD, enhancing the cycling power output.

## 1. Introduction

The popularity of the bicycle as an economical means of transportation and an effective tool for fitness and rehabilitation [1] justifies the number of biomechanical elements studied. Over the years, the biomechanical aspects of cycling that have been studied have included joint kinematics [2,3,4,5,6,7,8,9,10,11], kinetics [7,10,12,13,14], muscle activity using electromyography (EMG) [6,10,11,15,16,17], energy expenditure [2,18,19], effects of different workloads [20,21], cycling cadences [22,23], positioning of the subject on the bicycle [24,25,26], and performance of road racing/vibration behaviors [27,28]. All these elements are studied regarding performance, regardless of whether a cyclist’s goal is rehabilitation, recreation, or competition.

During cycling, the mechanical actions of the muscles are transmitted to the bicycle at different points of interaction, such as the handlebars, pedals, and seat. It is clear that the force applied to the handlebars and seat does not directly generate power, but these contact points address some concerns for investigation [29]. The biomechanics and efficiency of cycling are affected by seat height, crank arm length, and foot position [30]. The forces applied to the seat originated from the upper body weight, from the action of the arms on the handlebars, and from the reaction forces of the legs on the hip joint vary depending on the force produced by the legs [31]. When this force increases, there is an upwards force production (increased reaction forces at the hip joint) and the cyclist usually compensates for these forces by pulling on the handlebars and the pedals until they reach a point of a standing position [32].

Moreover, in relation to the upper limbs, while the cyclist is sitting on the bike, his arms act as force absorbers, resulting in the momentary imbalance typical of the pedaling gesture (cycling requires alternating right and left leg forces on the crank) [32]. In addition, considering the upper limbs and the contact with the handlebars, the force production that could be transferred to the lower limbs through the hip is only 3–5% of the crank power output [33]; however, the upper limb muscles facilitate reaching the highest power outputs, providing stable support for the action of the legs [34]. Regarding the handlebar height, some researchers have demonstrated that different handle heights change the rider’s trunk inclination and indirectly influence the stress on different parts of the rider’s body.

Sloane [35] and Delong [36] concluded that when riding on bikes with upward-bent handlebars, the upper body of the rider will be erected, shifting most of the body weight onto the saddle, thus compressing the intervertebral disks. Matheny [37] and Richmond [38] found that if the height of the handlebars is too low, it increases the probability of oppression on the nerves around the haunch, as well as symptoms caused by overuse, such as health problems involving the pudendal area in women and the prostate area in men. According to Chen and He [39], different handle heights caused various riding postures, resulting in different cervical and lumbar curvatures. The bicycles with higher handlebars were recommended when considering the spinal curvatures. These elements suggest that hand positioning may have a significant influence in the context of maximum power production [29]. Most bicycle adjustments are made to achieve a comfortable riding position and adequate range of motion in the lower extremities [40]. Therefore, geometric factors such as the handlebars height, as we have seen before, and the length of the bicycle frame, are generally adjusted to optimize the position of the bike seat.

Optimal bicycle rider position may be considered as a position in which force application and comfort are maximized, while resistive forces and risk of injury are minimized, in order to maximize bicycle velocity [41]. Concerning the bicycle frame, Ricard et al. [42] compared the effect of bicycle seat tubes on power production and EMG of four lower limbs muscles and verified that increasing the seat tube angle from 72° to 82° enabled triathletes to maintain power production, while significantly reducing the muscular activation of the bicep femoral muscles. Johnston [30] observed that the biomechanics and efficiency of cycling are affected by the seat height, crank arm length, and foot position. The point is to improve the comfort of cycling, bearing in mind the concept of “fitting an object to the human body” [43].

The cycling literature is filled with reports of EMG analyses for better understanding of the way muscles work when cycling, e.g., during pedaling exercises [32,44], regarding the variations of seat tube angle on muscle activation [45], according to the design of the bicycle frame [42], and during upright cycling [45,46], elliptical cycling [46,47], recumbent cycling [48,49], treadmill cycling [50], and uphill cycling [51]. Specifically considering the activity of the muscles of the upper limbs during cycling, we can observe studies related to: *lumbar erector spinae* [32,50,51], the *latissimus dorsi* (LD) [32,52], *the brachioradialis* [32,52,53], *biceps brachii* [32,51,52,53], *anterior deltoid* [32], *triceps lateralis* [32], *triceps brachii* [51,52,53], *upper trapezius* [53], *rectus abdominis* [51], *flexor carpi radialis*, *extensor digitorum* (ED), and *flexor digitorum* [32]. Although the previous studies provided useful knowledge concerning the muscular activity during cycling, the body of literature remained insufficient to understand the effect on muscular activity of different handlebar heights and bicycle frame lengths during cycling in order to support the work of researchers, coaches, and cyclists.

The muscular activity of the upper limbs in cycling is of relevant interest, and research in this area is scarce compared to that regarding the lower limbs, despite the fact that local fatigue contributes to global fatigue and influences sports performance in cycling. Hence, the purpose of this experimental study was to address the contribution of muscle activity, other than lower limbs, using different bicycle-rider systems (handlebar height; bicycle frame length). We hypothesized that (i) there are differences in muscular activation under different conditions, (ii) muscular activation patterns are similar using the same frame length, and (iii) the same handlebar height, and (iv) the increase in power generated leads to greater muscle activation, without changing the muscle activation patterns.

## 2. Materials and Methods

### 2.1. Subjects

Nine male recreational cyclists (age: 21.6 ± 1.9 years, body mass: 75.3 ± 5.8 kg; height 1.8 ± 0.1 m; crotch height: 9.5 ± 26.1 m; seat height: 0.7 ± 0.0 m; saddle distance from the handlebars, long: 0.6 ± 0.0 m; saddle distance from the handlebars, short: 0.5 ± 0.0 m, mean ± standard deviation) volunteered to participate in this study and were instructed to avoid strenuous exercise in the 24 h preceding each test session, to be well hydrated and fed, and to have abstained from caffeine and alcohol in the 3 h before each testing session. A recreationally active person was defined as someone who regularly participates in recreational activities for at least 30 min per day [54].

Subjects were recruited using a sample of convenience from the university campus, and the study involved a single-session research design. Prior to testing, all subjects completed and signed an informed consent form, stating the risks and benefits of the study. The ethics committee of the seeding institution (number 16019/2016) approved the procedures, which were in accordance with the Declaration of Helsinki of 1975, amended by the 64th WMA General Assembly, Fortaleza, Brazil, October 2013.

### 2.2. Instrumentation

Surface EMG signals from the anterior deltoid (AD), trapezius descending (upper) (TD), gluteus maximus (GM), and latissimus dorsi (LD) muscles on the right side of the body were measured.

Bipolar surface electrodes were used (10 mm diameter discs and 57 mm diameter with snap connector, Plux, Lisbon, Portugal), with an inter-electrode distance of 20 mm, and were placed in accordance with SENIAM recommendations [55]. The electrodes were positioned parallel to the muscle fiber orientation, with an interelectrode distance of approximately 2.0 cm. The skin was prepped by shaving, abrading, and cleaning with isopropyl alcohol prior to electrode placement. The ground lead was placed on the subject’s patellar tuberosity contralateral to the subject’s dominant limb [50]. The ground electrode was positioned over the cervical vertebrae. Transparent dressings with a label (Hydrofilm^®^, 10 × 12.5 cm, Rock Hill, SC, USA) were used to cover the electrodes and insulate them from perspiration. All cables were fixed to the skin by adhesive tape in several places to minimize their movement and, consequently, signal interferences.

All EMG was performed with MATLAB (Mathworks Inc., Natick, MA, USA) to determine muscle activity at neighboring points, where the energy was 30% of the maximum peak of muscle activation within a pedal stroke. The muscle activity was calculated by segmenting the muscle input signal energy according to the same criteria described by Stirn et al. [56].

Even though the high frequencies of the input signal were filtered with a Butterworth filter, muscle energy is very noisy and presents several local maximum peaks that do not correspond to the muscle active window center. To overcome this difficulty, the determination of the muscle’s “true” maximum energy peaks was carried out. Each pedal stroke performed by the cyclist produces patterns in the signal; these patterns are mainly translated by a periodicity in EMG energy. By determining the signal mean period, one can infer the maximum peak candidates using the highest and minimal differences between two maximum candidates and the expected period.

Once the maximum candidates were determined, the muscle activity boundaries were selected by finding the neighboring points where the energy was 30% of the determined maximum peaks. For each muscle activation, its active phase was defined as the part of the EMG signal for which the energy was at least 30% of the maximum local energy value for a given muscle activation. The raw EMG segments belonging to the active phases were extracted and used in the calculation of the duration of the active phase. The non-active phase was defined as the time interval between the two consecutive active phases.

EMG signals recorded during the selected times were filtered with a 4th order Butterworth filter (band width: 20–400 Hz), and the average rectified value (ARV) was calculated. ARVs at a specified speed were averaged every 7.3° of the crank angle because the encoder signal separated a crank cycle into 49 parts. The mean ARV during a crank cycle was calculated (total ARV) for analysis [57].

The temporal evolution of the mean durations of the active and non-active phases during the stroke were calculated for each muscle for the entire cycling time. Linear regression curves were fitted to the data, and the duration of the fitted curves at the time of the beginning and the end of the cycling session were compared.

For the assessment of muscle activity, only the dominant limb was selected in each subject.

### 2.3. Procedures

The anthropometric measurements, body mass, and crotch height of each participant were measured. The height (cm) was measured with a stadiometer (SECA, model 225, Hamburg, Germany) with a range scale of 0.10 cm. Weight and body mass were assessed using a Tanita body composition analyzer (model TBF-200, Tanita Corporation of America, Inc., Arlington Heights, IL, USA).

According to each participant’s anthropometric profile assessed before the test, the bicycle frame was measured by the seat-tube length and by multiplying the inseam measurement (floor to crotch in cm) by 0.66 [25]. The optimal seat height for cycling has been estimated by the LeMond method [58], by multiplying the inseam measurement (floor to crotch in cm) by 0.883 to get the distance from the center of the bottom bracket to the top of the seat. The handlebar height was established in two positions: (i) a high handlebar height, if it is 8 cm higher from the handlebars to the top of the seat; and (ii) a low handlebar height, if it is 8 cm lower from the handlebars to the top of the seat.

The test was carried out on a bicycle (Orbea, Road Racing Bike, Mallabia, Spain) equipped with a potentiometer (PowerTap, DT Swiss R460 Alloy) that allowed two handlebar positions and two frame lengths, supported on a roller (Tacx Sirius soft gel, Amsterdam, The Netherland).

To minimize the effect of circadian rhythms or differences in prior exercise, the same environmental conditions were applied to all tests, namely time of day (±2 h), temperature (28 °C), and relative humidity (50%).

Prior to the data collection, each subject performed a warm-up of five minutes of cycling in order to endorse a familiarization with the bicycle. After twenty minutes of passive rest, each cyclist performed the test session in the seated position, which lasted for approximately one hour and consisted of twelve different cycling conditions (Table 1).

### 2.4. Statistical Analysis

All data are shown as mean and standard deviation (SD). A general linear model (MANOVA) test was used after all the application assumptions had been ascertained to check the differences between muscle activation and the different test conditions. Tukey’s pos-hoc test was used to test changes with different intensities and to compare handlebar heights and bicycle frame lengths for total ARV, ARV during each order condition for each muscle, onset and offset of surface EMG, and cadence of cycling. Effect sizes were calculated using a partial Eta^2^ (*η*^2^). The level of significance was set at 0.05. Statistical analyses were performed using MATLAB (version 7, Math Works GK, Tokyo, Japan) and SPSS software (version 26.0, SPSS, Tokyo, Japan).

## 3. Results

The percentage of maximum voluntary contraction (% MVC) in the four studied conditions and muscles is shown in Table 2.

The MANOVA showed that the muscle activation of the different muscle groups studied differed, depending on the cycling condition (Pillai’s Trace = 2.487; F(36.69) = 9.300; *p* < 0.001, *η*^2^ = 0.958). Subsequent ANOVAs showed differences in muscle activation under the following cycling conditions: L1 [F(3.32) = 8.193; *p* < 0.001, *η*^2^ = 0.707]; L4 [F(3.32) = 5.733; *p* < 0.05, *η*^2^ = 0.400]; L5 [F(3.32) = 12.043; *p* < 0.001, *η*^2^ = 0.490]; L6 [F(3.32) = 284.817; *p* < 0.001, *η*^2^ = 0.784]; S7 [F(3.32) = 7.574; *p* < 0.001, *η*^2^ = 0.446]; S8 [F(3.32) = 122.875; *p* < 0.001, *η*^2^ = 0.712]; S9 [F(3.32) = 3.242; *p* < 0.05, *η*^2^ = 0.223]; S10 [F(3.32) = 5.874; *p* < 0.05, *η*^2^ = 0.329], and S12 [F(3.32) = 3.068; *p* < 0.05, *η*^2^ = 0.263].

Tukey’s post hoc analysis for an ARV electromyography activity was carried out, resulting in significant differences, as shown in Figure 1 and Figure 2 for ARV for both the long and short frames, respectively.

## 4. Discussion

Supported by the academic idea that the bicycle should fit the human body to maximize comfort [43] and performance [41], we demonstrated the contribution of muscle activity, other than lower limbs, using different bicycle-rider systems (handlebar height; bicycle frame length). For this purpose, we studied four muscles: AD, TD, GM, and LD, in four different conditions: HH, LH, LF, and SF, during three different intensities: 150 watts, 250 watts, and maximum intensity.

The study results highlight that, in general, there are differences in muscular activation under different conditions. Conversely, three exercise conditions showed no difference in the activation of the four muscles mentioned above: S11 (short frame and low handlebars at 150 watts) and L1 and L2 (high frame and high handlebars at 150 and 250 watts). These results are similar to the ones reported by Turpin et al. [32], who observed a lack of bursts of activity in AD and LD in the participant’s muscles while cycling at low intensity in the seated position. The absence of differences in muscle activity during low intensities allows us to declare that the most comfortable bicycle conditions are provided by the short frame with low handlebars, or the high frame with high handlebars.

However, muscular activation patterns are not similar using the same frame length or handlebar height. Our study identified several different patterns. For the short frame, we found a similar pattern between S9 (short frame and low handlebars at 250 watts) and S12 (short frame and high handlebars at 250 watts), with a weak muscle participation pattern and a slightly higher participation of GM. In contrast, the S8 exercise condition (short frame and high handlebars at 150 watts) was associated with a higher LD activation. The study by Hurst et al. [52] may explain these differences in muscle activation by clarifying that these differences may reflect differences in riding styles and body position. Simultaneously, concerning the long frame, the exercise condition L5 (long frame with low handlebars at 150 watts) showed that the predominant muscle is the TD, while in condition L6 (long frame with low handlebars at 250 watts), the predominant muscle is the GM. These results may indicate that muscle activity patterns change during cycling with varying cadences, as previously stated by Johnston [30].

Nevertheless, the most persistent pattern occurs during maximum intensity across all frame and handlebar conditions (L1—long frame and high handlebars; L4—long frame and low handlebars; S7—short frame and high handlebars; S10—short frame and low handlebars). In these conditions, the LD and AD presented the weakest muscular participation, while TD and GM presented higher muscular involvement than the other two. GM is the predominant muscle used during maximum intensity, except for the L1, where GM and TD are dominant. This specific muscle pattern may appear due to the body’s adjustment during high intensity cycling. The GM and TD activation suggest that the body switches from a comfortable position to a forward configuration, as Savelberg et al. [45] noted. This change in body position allows for an increase in power output through the pull of the handlebars and pedals [32].

Although studies indicate that LD may have contributed to the increase in acceleration [32], and is sensitive to the handlebar position, we did not find any indicator of a greater contribution of this muscle under the conditions studied [59]. This evidence reveals that muscle patterns, with the same handlebar position and frame, change at different intensities, which could have implications for trainers, athletes, and researchers in understanding what is the best bicycle position to reduce injuries and improve performance. In addition, this study allows for the adjustment of the handlebar position and frame at low intensities according to the patterns found, in order to prevent overtraining muscles.

There are some limitations to this study. First, the duration in each condition was short, so in the future, a more extensive analysis, with more time in each situation, will be needed to assess the effect of fatigue on these muscle patterns. Subsequent studies will allow us to understand whether the patterns found in this study will persist after longer cycling and if a specific handlebar height or frame length is more demanding for these particular muscles. Second, a subsequent study should be performed on women, as they can activate different muscle patterns while cycling under the same physical conditions [53]. Third, future studies should consider a larger sample size and the assessment of muscle symmetry, since this study only involved the dominant limb. Finally, our study results may differ from those obtained while cycling in the field, as we used a cycling roller. It would be helpful to recreate this study in the field [60]. In short, the study of muscle activation and patterns under different handlebar heights, frame lengths, and cycling intensities require further investigation.

## 5. Conclusions

Our study suggests that handlebar height and frame length influence muscle activation patterns during low cycling intensities, meaning that LD, AD, TD, and GM presented different contributions at low intensity using different handlebar heights and frame lengths. However, during high intensity, one specific pattern emerges, with a greater contribution of GM and TD and weaker participation of LD and AD, enhancing the cycling power output. This study highlights the importance of considering adjustments to details such as handlebar height and frame length when cycling under different intensities, with the aim of controlling fatigue levels in specific muscles with the goal of enhancing overall cycling performance.

## Figures and Tables

**Figure 1 ijerph-19-06590-f001:**
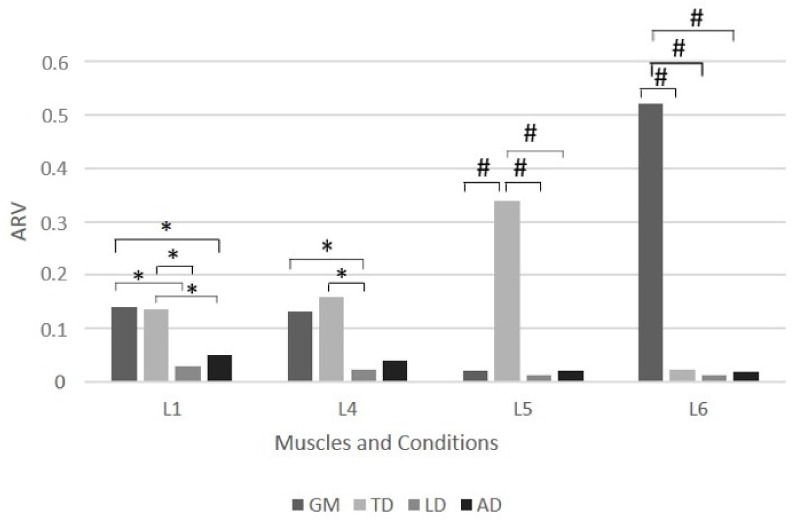
Differences between muscles according to ARV electromyographic activity (Tukey’s post-hoc test) in the different cycling conditions using a long bicycle frame. Differences between muscles for L1 (maximal intensity with high handlebars), L4 (maximal intensity with low handlebars), L5 (150 watts of intensity with low handlebars), and L6 (250 watts of intensity with low handlebars) conditions using a long bicycle frame. TD—trapezius descending; LD—latissimus dorsi; GM—gluteus maximus; AD—anterior deltoid. * Significance for *p* < 0.05; ^#^ Significance for *p* < 0.001.

**Figure 2 ijerph-19-06590-f002:**
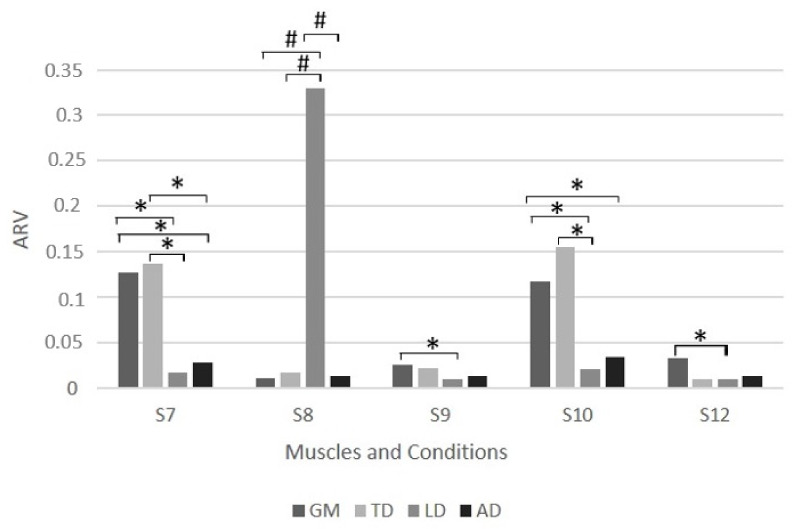
Differences according to ARV electromyographic activity (Tukey’s post-hoc test) in the different cycling conditions for a short bicycle frame. Differences between muscles for S7 (maximal intensity with high handlebars), S8 (150 watts of intensity with high handlebars), S9 (250 watts of intensity with high handlebars), S10 (maximal intensity with low handlebars), and S12 (250 watts of intensity with low handlebars) conditions for a short bicycle frame. TD—trapezius descending; LD—latissimus dorsi; GM—gluteus maximus; AD—anterior deltoid. * Significance for *p* < 0.05; ^#^ Significance for *p* < 0.001.

**Table 1 ijerph-19-06590-t001:** Test session protocol in twelve different cycling conditions.

Order Conditions	Bicycle Frame	Handlebar Height	Intensity	Time Duration	Resting	Abbreviation
1	Long (LF)	High (HH)	Max	30 s	4 min	L1 LF HH Max
2	Long (LF)	High (HH)	150 watts	1 min	2 min	L2 LF HH 150
3	Long (LF)	High (HH)	250 watts	1 min	3 min	L3 LF HH 250
4	Long (LF)	Low (LH)	Max	30 s	4 min	L4 LF LH Max
5	Long (LF)	Low (LH)	150 watts	1 min	2 min	L5 LF LH 150
6	Long (LF)	Low (LH)	250 watts	1 min	3 min	L6 LF LH 250
7	Short (SF)	High (HH)	Max	30 s	4 min	S7 SF HH Max
8	Short (SF)	High (HH)	150 watts	1 min	2 min	S8 SF HH 150
9	Short (SF)	High (HH)	250 watts	1 min	3 min	S9 SF HH 250
10	Short (SF)	Low (LH)	Max	30 s	4 min	S10 SF LH Max
11	Short (SF)	Low (LH)	150 watts	1 min	2 min	S11 SF LH 150
12	Short (SF)	Low (LH)	250 watts	1 min	3 min	S12 SF LH 250

Legend: LF—long frame; SF—short frame; HH—high handlebars; LH—low handlebars; min—minutes; s–seconds; Max—for maximal intensity.

**Table 2 ijerph-19-06590-t002:** Percentage of average rectified value for electromyographic activity of four muscles under different handlebar height and bicycle frame length conditions during cycling. Data expressed as percentage of maximum voluntary contraction (% MVC).

Cycling Conditions	HH LF			LH LF			HH SF			LH SF		
Intensity	max	150	250	max	150	250	max	150	250	max	150	250
Muscles	L1	L2	L3	L4	L5	L6	S7	S8	S9	S10	S11	S12
TD	9.20	1.34	13.71	10.95	29.76	1.24	10.36	1.15	1.64	12.36	4.39	4.90
(M ± *SD*)	*5.18*	*1.33*	*22.09*	*7.28*	*25.47*	*0.93*	*7.01*	*0.91*	*1.25*	*10.31*	*3.41*	*4.06*
LD	8.62	2.74	3.50	13.00	3.70	4.31	10.42	167.04	4.15	10.83	94.19	1.85
(M ± *SD*)	*4.87*	*0.56*	*1.44*	*10.59*	*0.51*	*1.58*	*3.60*	*79.90*	*2.10*	*3.62*	*105.91*	*0.71*
GM	13.80	1.06	3.79	12.73	1.45	44.88	12.95	1.00	3.05	11.79	30.24	5.49
(M ± *SD*)	*10.56*	*1.15*	*4.15*	*9.82*	*1.92*	*16.43*	*8.62*	*0.92*	*3.04*	*8.89*	*31.82*	*5.57*
AD	7.86	2.32	2.18	8.65	3.18	2.61	8.82	3.00	2.76	8.20	9.99	1.51
(M ± *SD*)	*4.53*	*1.75*	*1.60*	*5.24*	*1.89*	*1.28*	*5.85*	*1.50*	*1.21*	*5.94*	*3.31*	*0.88*

Legend: TD—trapezius descending; LD—latissimus dorsi; GM—gluteus maximus; AD—anterior deltoid. Exercise conditions: HH—high handlebars; LH—low handlebars; LF—long frame; SF—short frame. L1–L6—conditions with long frame; S7–S12—conditions with short frame; max—maximal intensity.

## Data Availability

The data that support the findings of this study are available from the corresponding author, upon reasonable request.

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
