# Peer review of "The Effect of Handlebar Height and Bicycle Frame Length on Muscular Activity during Cycling: A Pilot Study"

_ijerph, 2022, doi:10.3390/ijerph19116590_

Round 1
Reviewer 1 Report
Thank you allowing me to review again this study about cycling.
The authors have taken into account the comments previously made.
Although they took it into account in the discussion, I would have liked the unilateral side evaluation to be addressed a little more in the method section. However, the article is clear and merits further investigation.
Author Response
Thank you allowing me to review again this study about cycling.
Thank you for your willingness to continue to make this article better.
The authors have taken into account the comments previously made.
We feel that the comments were all quite relevant and taking them all into consideration we feel the article is substantially better.
Although they took it into account in the discussion, I would have liked the unilateral side evaluation to be addressed a little more in the method section. However, the article is clear and merits further investigation.
Thank you very much for the comment
We took this into account and added the paragraph in the method section: "For the assessment of muscle activity, only the dominant limb was selected in each subject".
English review
In order to improve English, we sent the article to an editorial (attached certificate)
“This document certifies that the manuscript listed below was edited for proper English language, grammar, punctuation, spelling, and overall style by one or more of the qualified proficient English editors at Silabas Didaticas Unipessoal.”

Reviewer 2 Report
The authors correctly addressed the previous comments, and now I believe the manuscript can be published.
I would recommend reworking the plots (Figures 1 and 2) because the quality seems poor both in terms of image quality and data representation. (like all these symbols ## #* etc.).
Author Response
The authors correctly addressed the previous comments, and now I believe the manuscript can be published.
We feel that the comments were all quite relevant and taking them all into consideration we feel the article is substantially better.
I would recommend reworking the plots (Figures 1 and 2) because the quality seems poor both in terms of image quality and data representation. (like all these symbols ## #* etc.).
We made the graphics again, increasing their quality, using a specific program (GRAPHPAD), and as proposed we changed the previous symbols.
Thanks for the comment, we now consider that the graphics are more noticeable.
ENGLISH REVIEW
In order to improve English, we sent the article to an editorial (attached certificate)
“This document certifies that the manuscript listed below was edited for proper English language, grammar, punctuation, spelling, and overall style by one or more of the qualified proficient English editors at Silabas Didaticas Unipessoal.”

This manuscript is a resubmission of an earlier submission. The following is a list of the peer review reports and author responses from that submission.
Round 1
Reviewer 1 Report
This work shows the effect of handlebar height and frame length on the activation of some muscles demanded during cycling. I found the work well written, with correctly applied methods and an adequate discussion of the results. I have a few comments below.
I missed the inclusion of standard deviations in the results. It is important to understand the variability of tests.
I recommend not including the title ("Average rectified value for...") in the graphs, as this information is in the legend caption. The explanatory text ("Difference between muscles...") should also be in the legend caption.
Reviewer 2 Report
Thank you allowing me to review this study about cycling.
Have few major remarks:
Firstly, I think authors should better explicit the interest of their study and what they aimed to show. Indeed, I don't really see the clinical practice and interest of your results. Authors should explain why their study is valuable, especially in their conclusions.
The second issue is the choice of studying only the right side. Indeed, I think it would have been interesting to compare both sides and analyze muscular symmetry. At least authors should discuss this main limitation in their "limits".
Minor:
Please, revise anthropometrics: sometimes cm for m and conversely (l 114-115).
Table 2 legend: please write in full ARV
Reviewer 3 Report
The manuscript titled “The Effect of Handlebar Height and Bicycle Frame Length in Muscular Activity During Cycling” investigates the EMG activity of back’s muscles in different bicycle-rider systems during a cycling session. The aim is interesting since the potentiometer may be included in the bicycle manufacture, however several concerns may prejudice the manuscript publication.
The aim and the contents are acceptable, the authors correctly approached the scope and provided enough information about the aim. The introduction is too long, I suggest reducing it drastically. Abbreviations are not required if the specific term is not repeated frequently (L95-99). Furthermore, lines 105-110 appear like the conclusion of the study. Authors write for the first time the muscles involved in the study in L128-130 with the respective abbreviations; however, in the Discussion, the authors reported the full-length name again and added the abbreviation again.
Besides these suggestions, the sample size (nine participants) is too small, not enough representative of the population. They are nine young students aged 20-22 years, all the same height; too homogeneous to represent any population; additionally, there are no statistical tests to assess the validity of the sample. Due to this, I believe that the manuscript cannot be published in this actual form. However, the authors could re-submit it in a “pilot-study” version since nine participants would be sufficient for this case.